# Simulation of the Effect of Correlated Packet Loss for sUAS Platforms Operating in Non-Line-of-Sight Indoor Environments

**Edwin Meriaux [1],\*, Jay Weitzen [1] and Adam Norton [2]**

1   Lowell ECE Department, University of Massachusetts, Lowell, MA 01854, USA; jay_weitzen@uml.edu
2   Lowell NERVE Center, University of Massachusetts, Lowell, MA 01852, USA; adam_norton@uml.edu
\*   Correspondence: edwin.meriaux@mail.mcgill.ca

**Abstract:** The current state of the art in small Unmanned Aerial System (sUAS) testing and evaluation exists mainly in the realm of outdoor flight. Operating small flying sUAS in constrained indoor or subterranean environments places different constraints on their communication links (control links and camera/sensor links). Communication loss in these environments is much more severe due to the proximity of obstacles. This paper examines how correlated packet loss (burst errors) occurring on both the control and camera communication links affects the ability of pilots to fly and navigate small sUAS platforms in constrained Non-Line of Sight (NLOS) environments. A software test bench called AirSim, a UAV simulator, allows us to better understand the effects of correlated packet loss on flyability without damaging multiple sUAS units by flight testing. The simulation was designed to support the design of test methodologies for evaluating the robustness of the communication links and to understand performance without damaging flight tests. Throughout the simulations, it is observed how different levels of packet loss affect the pilot and the number of simulated crashes into the obstacles placed through space. The simulations modeled packet loss both on the video link and the control link to display how packet loss affects ability to pilot and control the sUAS. The utility of using a simulated environment rather than flight testing prevents damage to the fragile and expensive drones being used.

**Keywords:** sUAS; packet loss; correlated packet loss; non-line of sight; AirSim; FPV; NLOS; BLOS; subterranean

## 1. Introduction

Standard sUAS platforms, also called flying drones, are mostly designed for operation outdoors [1,2], most often operating in a direct line of sight between the controller and the sUAS and using GPS for navigation. There are also sUAS built for operation in underground or indoor environments, but they are less common. For example, most drones use GPS to navigate, but this is unavailable in indoor or underground environments. Communication, in general, is more difficult in indoor and subterranean cases. The work described in this paper is part of a larger project [3,4] to develop test methods to evaluate and compare how different drones operate in indoor and subterranean environments, such as mines [5] and sewers [6]. The capabilities tested included communications [7], navigation [8], obstacle avoidance, mapping [9], human-robot trust [10], and autonomy [11]. The communication function of the testing is critical. This simulation work is specifically built as an extension to flight testing for communication that has already been conducted for this project. There are two goals for this paper. The first is to be able to run simulated communication testing, and the second is to obtain the results from simulation testing and understand the results from the simulation data. The usefulness of these virtual benchmarks is that no drones are damaged in flight testing. The drones used in the flight testing for this project cost tens of thousands of dollars and could be easily damaged by collisions. This is particularly common when communication loss tests are run. When this occurs, the drone can be

rendered effectively useless unless it has significant autonomy. During periods of degraded or lost communication, it might collide into walls or other objects in the space [12,13]. To better understand how drones are affected by degraded communications without risking potentially damaging flight tests, some critical information needs to be understood about how they behave under different communications loss scenarios.

Most sUAS platforms use one of three standard bands for control and data communication (camera): 1.8 GHz (military band in the US), 2.4 GHz ISM (80 MHz unlicensed), and 5.8 GHz unlicensed bands (125–250 MHz). In free space, from the Friss Free Space Equation, every doubling of the frequency results in approximately 6 dB additional path loss, everything else being equal. Therefore, going from 2.5 GHz to 5 GHz results in approximately 6 dB additional path loss. 1.8 GHz vs. 2.4 GHz is about 2.5 dB in additional loss in free space. In addition to attenuation in the non-Line-of-Sight (NLOS) environment, many of these units operate in unlicensed bands within which Wi-Fi and other unintentional interference can elevate the noise floor and degrade the usable range. This is before considering the effects of multipath and fading, which are present in indoor environments.

Most of the units tested use separate control and data communications streams. Because the video camera feed tends to have a much higher bandwidth than the control channel, the loss of the video feed often occurs before the loss of the control channel. Another feature of the drone control software is that when a control packet is lost, most units continue using the last correctly received control velocity trajectory instruction until it regains contact.

When operating non-line of sight (NLOS) [14–17], the pilot flies using what they see from the camera and other on-board sensors communicated via a telemetry link. Packet loss on either the control link or the telemetry link can introduce latency or control degradation into the piloting system. Reference [18] shows how packet loss can affect drones in larger spaces while moving along a trajectory. In that situation, variations in the flight due to different types of packet loss can be seen. However, there is not as much emphasis on the environment, so there are not the same risks as in constrained environments. In constrained environments, any variations in flight path can result in the unit hitting walls, losing control, or being damaged. Subterranean environments have an extra level of difficulty associated with their environment, especially with correlated packet loss and the increased danger, and this will be tested in this paper.

This paper develops a virtual series of tests to understand the effect of correlated packet loss (bursts of lost packets) on sUAS platform's communication link. This loss affects the pilot and causes deviations from the intended flight paths, mainly from the view of the pilot. The simulator can also be used to see how different possible test methods can be implemented for drones by testing in simulation first. Many different tests will be run. The first set of experiments examine how the effect of communication loss affects a drone that is not being flown by a person (autonomous control from a centralized point) and the second set of tests requires a pilot flying. The pilot will have both video feed and communication loss. The loss will be tested separately and combined. The results combined with the flight testing from previous work show that communications testing can be run in simulation to understand the effect from packet loss on drones.

## 2. Methodology for Evaluating the Effect of Packet Loss on sUAS Piloting

The indoor wireless environment is characterized by multiple reflections and relatively low velocities (small Doppler spreads), which means that when a drone is in a fade, it will likely result in multiple data/control packets being lost. The packet layer is the concentration of this paper since this communication channel, which we model as a 2 state Gilbert-Elliot process [19–21] shown in Figure 1, in which it is assumed that either the channel is good or bad, with probabilities $P_g$ and $P_b$, respectively. When the channel is bad, one assumes that a burst error of $N_b$ consecutive packets occurs, where $N_b$ is Poisson distributed. Given the modulation and coding that is used in most modern sUAS systems,

the Signal to Noise Ratio (SNR) vs. Error Rate curves are so steep that the Gilbert-Elliot process represents a good baseline model for the channel at the packet layer.

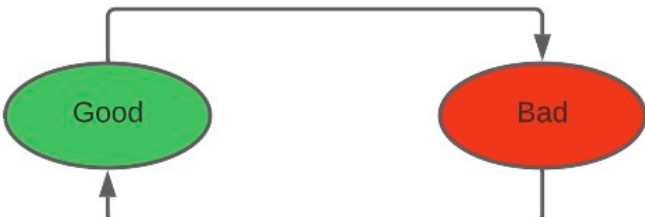

**Figure 1.** A Two-State Gilbert-Elliot Channel Packet Loss Model.

To examine how much packet loss can be tolerated before the sUAS becomes hard to pilot in a constrained environment (pilots report controls become "sluggish") or it hits a wall, a simulation was designed using AirSim simulation platform [22]. AirSim was designed by Microsoft to provide an environment for simulating flying drones [23,24]. AirSim has been used in many underground testing scenarios in the past to train and test drones before flying in real tunnels. The DARPA Subterranean Challenge is an example of this [25], and it has led to other underground projects being conducted in this simulator [26]. The AirSim code was modified, and new routines were developed to simulate the effects of correlated packet loss. This was built using the Robot Operating System (ROS) to apply publishers and subscribers, which effectively act like a User Datagram Protocol (UDP) connection that drones use (similar to Wi-Fi connections). This was for the communication and video feeds of the drones. Control and video packets were sent at a rate of 10 packets per second and generated loss events with a fixed duration $N_b$ packets with a probability $P_b$. The sUAS moved at a constant velocity of V m/s, which means that during an outage lasting $T_o = Nb/10$ s, the unit will move $VT_o$ meters using the last correctly received packet. If the unit trajectory is moving in a straight line, loss of packets may not be critical, but if the unit must be maneuvered, then packet loss can be critical. The ways packets can be lost are divided into a few categories. The first set is whether the packet loss is correlated or not. The second is which communication packet is lost, control, video, or combined control and video loss. The packet loss was modelled as dropping packet on the subscriber. When the subscriber did not receive a communication or video feed packet, it used the last packet received. In real life, not all drones always do this. When the packet loss is too severe, they sometimes automatically land (ending the useful mission) while other drones continue using the last received packet. This was validated with flight testing of different units.

Tables 1 and 2 from reference [7] present flight test data showing the effect of communication degradation due to obstacles degrading the drone's controller feed and video feed. In Table 1, as the number of walls and obstacles increases, the rate at which the good packets arrive decreases to the controller. Table 2 shows the video feed data are equally affected in the vertical and horizontal with packet loss from obstacles. With simple walls made from drywall, the packet loss is not too severe. This is in contrast with concrete, which effectively blocks the communication, resulting in near 100% packet loss. These flight tests were conducted in simple cases where the drone was not flying complicated missions. The experiments were designed to test the packet loss. Now that communication issues are better understood with the empirical flight test data, the environment can be ported into simulation. Losses modeled are control/telemetry channel and video feed loss. The simulator shows how the increase in packet losses affects the system and ability to pilot. The utility of modeling drones in a simulator and testing how they react to packet allows the drone to be evaluated without damaging the drone. In terms of the development of the drone, it can be useful to test before the drone is built so that rapid prototyping can be done without the need for the construction of each iteration of the drone.

**Table 1.** NLOS Operations Data Table Controller Loss (// indicates Poor Quality, ✓ indicates good quality, X indicates no communication, green highlighting indicates Pass, red highlighting indicates Fail).

| sUAS, Communications Frequency, and OCU Signal Indication | Metrics | Horizontal, through Walls | | | | | | Vertical, through Floors | | | | |
|---|---|---|---|---|---|---|---|---|---|---|---|---|
| | | X | 1 | 2 | 3 | 4 | 5 | X | 1 | 2 | 3 | 4 |
| Example Unit A 2.4 GHz | Video | ✓ | ✓ | ✓ | // | // | X | ✓ | // | // | X | X |
| | Control | ✓ | ✓ | ✓ | ✓ | ✓ | X | ✓ | ✓ | X | X | X |
| | Takeoff | ✓ | ✓ | ✓ | ✓ | ✓ | X | ✓ | ✓ | X | X | X |
| | Hovering | ✓ | ✓ | ✓ | ✓ | ✓ | X | ✓ | ✓ | X | X | X |
| | Yawing | ✓ | ✓ | ✓ | ✓ | ✓ | X | ✓ | ✓ | X | X | X |
| | Pitching | ✓ | ✓ | ✓ | ✓ | ✓ | X | ✓ | ✓ | X | X | X |
| | Rolling | ✓ | ✓ | ✓ | ✓ | ✓ | X | ✓ | ✓ | X | X | X |
| | Ascend and descend | ✓ | ✓ | ✓ | ✓ | ✓ | X | ✓ | ✓ | X | X | X |
| | Camera movement | ✓ | ✓ | ✓ | ✓ | ✓ | X | ✓ | ✓ | X | X | X |
| | Landing | ✓ | ✓ | ✓ | ✓ | ✓ | X | ✓ | ✓ | X | X | X |
| | Maximum NLOS performance | 27 m, 4 walls | | | | | | 5 m, 1 floor | | | | |
| Example unit B 1.8 GHz | Video | ✓ | // | // | X | X | X | ✓ | // | X | X | X |
| | Control | ✓ | ✓ | ✓ | X | X | X | ✓ | ✓ | X | X | X |
| | Takeoff | ✓ | ✓ | ✓ | X | X | X | ✓ | ✓ | X | X | X |
| | Hovering | ✓ | ✓ | ✓ | X | X | X | ✓ | ✓ | X | X | X |
| | Yawing | ✓ | ✓ | ✓ | X | X | X | ✓ | ✓ | X | X | X |
| | Pitching | ✓ | ✓ | ✓ | X | X | X | ✓ | ✓ | X | X | X |
| | Rolling | ✓ | ✓ | ✓ | X | X | X | ✓ | ✓ | X | X | X |
| | Ascend and descend | ✓ | ✓ | ✓ | X | X | X | ✓ | ✓ | X | X | X |
| | Camera movement | ✓ | ✓ | ✓ | X | X | X | ✓ | ✓ | X | X | X |
| | Landing | ✓ | ✓ | ✓ | X | X | X | ✓ | ✓ | X | X | X |
| | Maximum NLOS performance | 25 m, 3 walls | | | | | | 5 m, 1 floor | | | | |

**Table 2.** Video Latency Data Table (X indicates no communication, green highlighting indicates Pass, red highlighting indicates Fail, grey indicates cases in the averages that cannot be calculated due to failed tests).

| sUAS and Communications Frequency | Trial | Video Latency (ms) | | | | | | | | | | |
| --- | --- | --- | --- | --- | --- | --- | --- | --- | --- | --- | --- | --- |
| | | Horizontal, through Walls | | | | | | Vertical, through Floors | | | | |
| | | X | 1 | 2 | 3 | 4 | 5 | X | 1 | 2 | 3 | 4 |
| Example A 2.4 GHz | 1 | 33 | 33 | 35 | 37 | 39 | X | 33 | 37 | X | X | X |
| | 2 | 37 | 37 | 39 | 40 | 42 | X | 37 | 40 | X | X | X |
| | 3 | 33 | 33 | 35 | 37 | 39 | X | 33 | 37 | X | X | X |
| | 4 | 50 | 50 | 53 | 55 | 58 | X | 50 | 55 | X | X | X |
| | 5 | 37 | 37 | 39 | 40 | 42 | X | 37 | 40 | X | X | X |
| | 6 | 33 | 33 | 35 | 37 | 39 | X | 33 | 37 | X | X | X |
| | 7 | 33 | 33 | 35 | 37 | 39 | X | 33 | 37 | X | X | X |
| | 8 | 37 | 37 | 39 | 40 | 42 | X | 37 | 40 | X | X | X |
| | 9 | 33 | 33 | 35 | 37 | 39 | X | 33 | 37 | X | X | X |
| | 10 | 50 | 50 | 53 | 55 | 58 | X | 50 | 55 | X | X | X |
| | Average video latency | 38 ($\pm$ 7) | 38 ($\pm$7) | 40 ($\pm$7) | 42 ($\pm$7) | 44 ($\pm$8) | n/a | 38 ($\pm$7) | 42 ($\pm$7) | n/a | n/a | n/a |
| | Latency at maximum NLOS range | 44 ms ($\pm$8 ms) 27 m, 4 walls | | | | | | 42 ms ($\pm$7 ms) 5 m, 1 floor | | | | |
| Example B 1.8 GHz | 1 | 17 | 18 | 20 | X | X | X | 17 | 22 | X | X | X |
| | 2 | 17 | 18 | 20 | X | X | X | 17 | 22 | X | X | X |
| | 3 | 17 | 18 | 20 | X | X | X | 17 | 22 | X | X | X |
| | 4 | 23 | 26 | 28 | X | X | X | 23 | 31 | X | X | X |
| | 5 | 17 | 18 | 20 | X | X | X | 17 | 22 | X | X | X |
| | 6 | 20 | 22 | 24 | X | X | X | 20 | 27 | X | X | X |
| | 7 | 17 | 18 | 20 | X | X | X | 17 | 22 | X | X | X |
| | 8 | 17 | 18 | 20 | X | X | X | 17 | 22 | X | X | X |
| | 9 | 17 | 18 | 20 | X | X | X | 17 | 22 | X | X | X |
| | 10 | 23 | 26 | 28 | X | X | X | 23 | 31 | X | X | X |
| | Average video latency | 18 ($\pm$3) | 20 ($\pm$3) | 22 ($\pm$3) | n/a | n/a | n/a | 18 ($\pm$3) | 24 ($\pm$4) | n/a | n/a | n/a |
| | Latency at maximum NLOS range | 22 ms ($\pm$3 ms) 25 m, 3 walls | | | | | | 24 ms ($\pm$4 ms) 5 m, 1 floor | | | | |

## 3. Test Environment

The AirSim simulator allows for customized environments in the "Unreal" gaming engine, (a standard game engine used for many video games and CGI for movies) and with modular Python programs it can be used to emulate a drone flight. The flight controller used in AirSim is the PX4. This is a very common controller used in the industry. During flight testing in this project, all the units used PX4 controllers. This implies that in future tests, instead of the generic PX4 being used, the actual software of the desired drone can be emulated. This provides more accurate testing before a drone is ever flown.

In this simulation ROS, was used to model communication using publishers and subscribers to mimic the controller and video packets being sent back and forth during flight. Two environments were designed with different tests run in each. These were used to help design the test methods to compare real sUAS units.

This paper is focused on non-Line of Sight drone operation, since communication loss will not be as much of a factor in Line-of-Sight situations. The two sub-cases of non-Line of Sight depend on how the drone is flown. If it is flown in an autonomous manner, its communication loss with its central control center becomes important. If the drone is flown in "First Person View" (FPV) manner, then both its communication loss and video feed loss

become important. These effects were seen in the flight testing when non-Line-of-Sight autonomous testing was conducting the packet loss was different than when it was being flown by a user (Line of Sight) (see Tables 1 and 2). These two cases were built in simulation.

*3.1. Test Environment 1: Circular Path*

The first environment was the simple default AirSim empty environment (Figure 2a). In this case, the drone flies around in a wide circle in a large space with no obstructions. The flight of the drone was based on a predetermined path. At each time step, it was programmed to the next point on the circle (Figure 3a represents this path). The controller used in this test represents what a real drone would use, so it is split up across two Python processes (one for the controller and one for the drone). There is no video communication in this test, since everything is based on using predetermined points. The controller knows where the drone currently is and where the drone needs to be next, so it publishes the necessary X/Y/Z speeds for the drone over ROS.

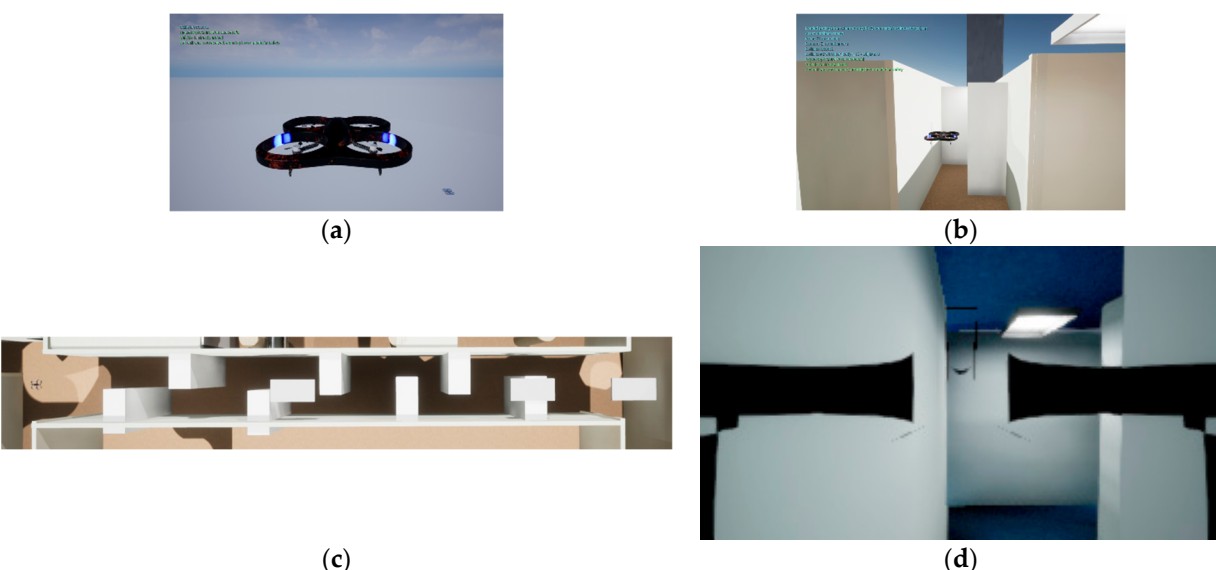

(a)    (b)

(c)    (d)

**Figure 2.** (**a**,**b**) Airsim Test Course Environment 1, Start View (**left**) Test Course Environment 2, Start View (**right**). (**c**) Airsim Test Course Environment 2, Top View. (**d**) AirSim Test Course Environment 2 FPV, Drone View.

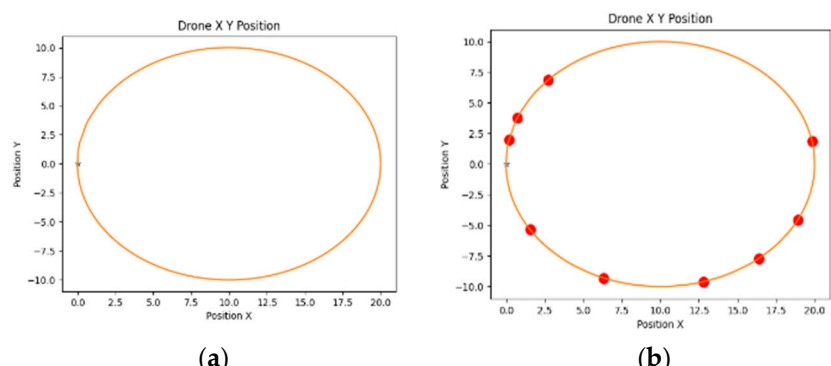

(a)    (b)

**Figure 3.** (**a**) is the ideal flight path with no packet loss. (**b**) is a case with 5% loss and packet loss events only last 1 packet (0.1 s).

The other process subscribes to this, and by receiving this data, the drone can follow the new path. Packet loss is simulated on the subscriber side by ignoring the newest packet and following the previous packet. In correlated loss, it follows the original packet before any loss occurred. As more packets are lost, the drone will deviate more from

its intended path. This means the average deviation from path will increase. Since the controller operates at 10 packets per second, a 1 s fade means the loss of 10 consecutive control packets. A circular path was chosen because constant course alteration is required that allows observation of the effect of delays in correcting the path. Path deviation is used as a metric in this test due to the fact that the drone is intended to follow a direct path and any deviation from that is due to packet loss. When there is no loss, the deviation is near zero. This space is free of obstacles, so collision count is not a valid metric.

### 3.2. Test Environment 2: Hallway Obstacle Course

The second environment was modelled after a lab space we used at the University of Massachusetts Lowell for flight testing drones. This 1:1 replica provides flexibility in the path the drone must take from start to end (Figure 2b,c). The hallway was filled with blockages that required the drone to maneuver around them. The simulated controller module was modified to allow a user to control the drone instead of just outputting the predetermined path. A key logger was added into the original code to allow for keyboard commands (but this can be easily modified to allow a game controller instead). The same type of error, as in the controller for test environment 1, was implemented, allowing for controller communication errors. But, in this case, it is important since depending on what the user sees, they can fly differently. This means that video loss error was also added separately and in combination with the control loss.

In AirSim, by default, the user has a more omniscient view, but for the sake of these tests, a FPV setup was configured (Figure 2d). This is the way many drones are piloted, especially in NLOS environments. Errors in the video link were incorporated to simulate packet loss on the video feed. Again, just like on the controller, correlated packet loss was incorporated into the simulation. Since the controller operates at 10 packets per second, a correlated loss of 10 packets means the loss of a full second of data. In the video feed, the frame rate was around 10 FPS, so losing 10 packets in a row also represents 1 s. Collision count is used in these experiments. Path deviation is not a good metric because since a user is flying it, no human can perfectly fly the same path consistently.

## 4. Simulation Results

### 4.1. Test Environment 1: Circular Path

This scenario requires the drone to fly in a circle. Figure 3a shows the flight with no loss. Figure 3b simulates occasional single packet losses of 5%. As presented in the figure, it is barely noticeable, as summarized in Table 3. Aside from the 50% packet loss case, no drone with one packet loss per drop event has a much higher RMS error than the rest.

**Table 3.** RMS error versus Pb and Nb.

| Packet Loss | RMS: Error 1 Packet Loss per Drop Event | RMS: Error 2 Packet Loss per Drop Event | RMS: Error 5 Packet Loss per Drop Event |
|---|---|---|---|
| 0.00% | 0.319 | 0.320 | 0.3194 |
| 5.00% | 0.319 | 0.320 | 0.627 |
| 10.00% | 0.320 | 0.454 | 22.649 |
| 15.00% | 0.320 | 0.524 | 25.870 |
| 20.00% | 0.322 | 0.370 | 59.2430 |
| 25.00% | 0.321 | 13.706 | 55.639 |
| 50.00% | 24.535 | 65.924 | 69.093 |

When correlated burst packet loss is introduced, it is observed that the course becomes less than circular, as the auto-pilot struggles to maintain the proper course. This corresponds to reports of "sluggish controls" reported by pilots during experiments indoors near the edge of coverage. This is illustrated in Figure 4a,b. When multiple packets are lost in a row, there can be increasing errors in the flight path. This can be seen in more detail in Table 3, where the different packet loss percentages and correlated losses are shown. Just an

increase in the percentage of random packet loss is not enough to cause an increase in RMS error until 50% packet loss is caused. On the other hand, as correlated packet loss increases, the RMS error starts to increase immediately. The simulated data collected makes intuitive sense because with more error, a drone deviates more from its path, and therefore the RMS error increases. The 50% packet loss case was an extreme case, showing how un-flyable it became. In an underground or indoor case, the increase in path deviation will cause an increase in collisions with walls and obstacles. This data lines up with the results from the flight tests with the results shown in Table 1 for controller loss. As the number of walls between the drone and the control center increases, the drone becomes less and less flyable. When the drone's communication is almost cut out due to being behind obstacles, as seen in Table 1, the drone cannot be flown. This is modeled in the simulator with an increase in loss. The fact that this lines up indicates that the flight test results are validated, meaning that some simulation work can become simulated to reduce the cost of flying a real drone and its cost of repair if damaged.

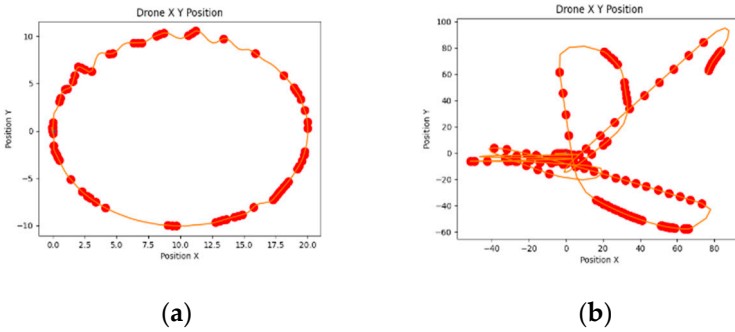

(**a**)　　　　　　　　　　　　　　　　　　　　(**b**)

**Figure 4.** sUAS course under high burst packet loss conditions: (**a**), 20% packet loss change, and every packet loss event lasts 1 packet (0.1 s). In (**b**), 50% packet loss changes and every packet loss event lasts 10 packets (1 s).

### *4.2. Test Environment 2: Indoor Path*

The second simulation environment corresponds to an indoor obstacle course in which the unit traverses a straight line toward the goal point while making a series of maneuvers when it approaches the obstacles shown in Figure 2b,c. In this case, the unit is piloted by a user seeing the drone's FPV view (Figure 2d).

This set of tests breaks down into three cases: only controller packet loss, only video packet loss, and combined packet loss. An example of a typical experiment is seen in Figure 5a–d. In the figures, a red dot represents a collision between the sUAS and a wall or obstacle. The reason the collisions are focused on the corners is because it is relatively easy to fly in straight lines with packet loss, but it is much more difficult to turn with packet loss included. In this set of simulations, instead of RMS path error, the metric used to analyze the effect of packet losses was the number of collisions with a wall during the flight. In our simulations, the drone being tested is assumed to have propeller guards so that an encounter with a wall does not damage either the wall or the drone. The drone is set so that collisions will also not cause it to tilt. When drone collides with the wall, there is risk of damage or falling, terminating the flight, even with propeller guards. This is backed up by the flight testing during this project.

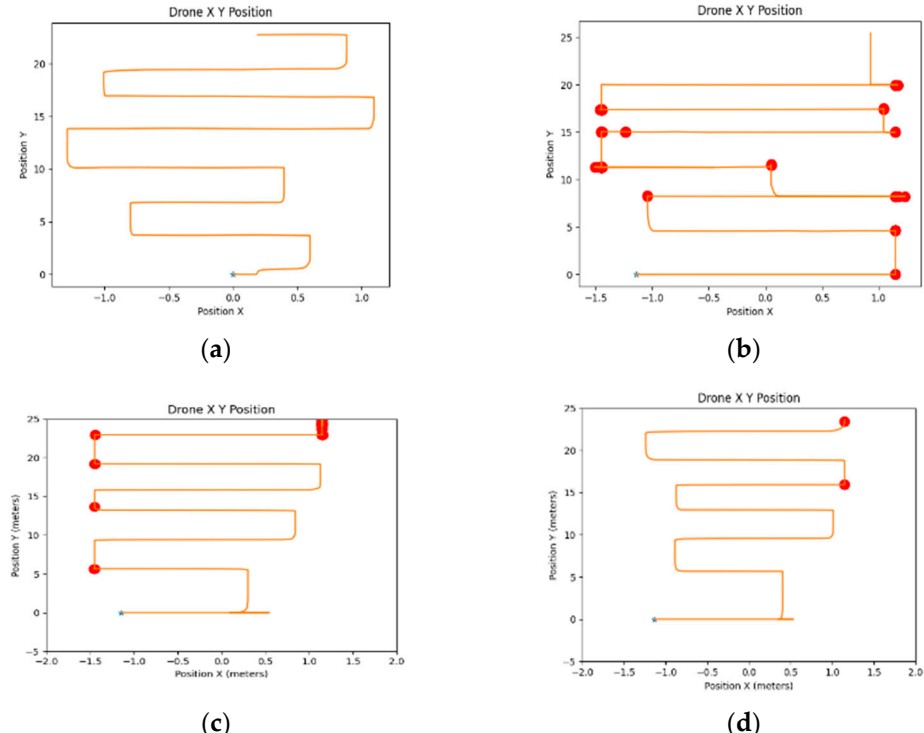

**Figure 5.** (**a**) A typical flight path with no wall touches and no packet loss. (**b**) A 25% controller packet error with 10 packets lost at each loss event rate showing wall touches with red dots. (**c**) A 10% controller packet error with 5 packets lost at each loss event (0.5 s) rate showing wall touches with red dots. (**d**) A 10% controller and video packet error with 5 packets lost at each loss event (0.5 s).

In Figure 5a–d, different types of packet loss combinations are shown. The data for this can be seen in Tables 4–6. The tests for the control packet loss were run at multiple packet loss rates and at multiple packets dropped per loss (correlated loss). Each test case is the average collision number over five runs.

**Table 4.** Controller loss with different packet losses and drops per event. Grid shows collision based on the data.

| Controller Loss Collision Count | | | | | | |
|---|---|---|---|---|---|---|
| **Drops per Event** | **Average Percentage Packet Loss** | | | | | |
| | 0 | 5 | 10 | 15 | 20 | 25 |
| 0 | 0 | 0 | 0 | 0 | 0 | 0 |
| 1 | 0 | 2.2 | 2.8 | 0.6 | 1.8 | 3 |
| 5 | 0 | 2.6 | 4.8 | 5.2 | 7 | 8.8 |
| 10 | 0 | 5.4 | 7 | 11.6 | 12.6 | 20 |

**Table 5.** Video loss with different packet loss percentages and drops per event. The grid shows collisions based on the data. N/A indicates tests in which there was so much packet loss that the tests were not completable.

| Video Loss Collision Count | | | | | | |
|---|---|---|---|---|---|---|
| **Drops per Event** | **Average Percentage Packet Loss** | | | | | |
| | 0 | 5 | 10 | 15 | 20 | 25 |
| 0 | 0 | 0 | 0 | 0 | 0 | 0 |
| 1 | 0 | 0.8 | 0.2 | 1.4 | 1.4 | 2.4 |
| 5 | 0 | 0.6 | 4.6 | 10.6 | N/A | N/A |
| 10 | 0 | 10.6 | N/A | N/A | N/A | N/A |

**Table 6.** Combined controller and video loss with different packet losses. The grid shows collisions based on the data. N/A indicates tests in which there was so much packet loss that the tests were not completed.

| | Controller and Video Loss Collision Count | | | | | |
|---|---|---|---|---|---|---|
| **Drops per Event** | **Average Percentage Packet Loss** | | | | | |
| | 0 | 5 | 10 | 15 | 20 | 25 |
| 0 | 0 | 0 | 0 | 0 | 0 | 0 |
| 1 | 0 | 0.8 | 0.6 | 1.6 | 1.8 | 4.8 |
| 5 | 0 | 2.8 | 6.2 | 17.8 | N/A | N/A |
| 10 | 0 | 6.2 | N/A | N/A | N/A | N/A |

Once video loss is added into the experiment (Tables 1 and 2), there are cases where the drone is so badly affected by packet loss that there is no way to complete the mission. This means there is so much loss that the drone is just constantly colliding into the wall, and after about 10 min of flight, the drone is still not at its destination (in a no-loss situation the drone should only take 1 min to arrive at its goal).

Tables 4–6 data are relatively intuitive. With a higher packet drop percentage and higher drop number per event, the collision rate increases. The only exception is at 1 packet drop per loss, which is the random case. This lines up with the data from Table 3 in the first experiment. Low packet drops per loss, even with higher packet loss percentages, does not really affect the drones. The interesting result comes from comparing the results of those two tables (Tables 4 and 5) together (meaning Table 6). Individually, they show how different packet losses affect the drones differently. With correlated packet loss of 5 or 10 drops per event, the video loss (Table 5) tests have a greater collision number than the case with only controller loss (Table 4). Table 6 displays that the drone is significantly affected by the combined packet loss, as the collision rate is at its highest. The reason video loss affects the pilot more could be because flying blind is more complicated than flying with lag in the control. In the controller loss cases, the drone is just continuing along the original path.

Under communication loss, regardless of the action the pilot wants it to do, the drone is not receiving the commands. Since the path in this test was mainly linear, if the loss happened at any point, except where a turn is needed, there will not be a collision. Video loss is more complicated to deal with because the pilot does not know exactly where the drone is because of loss and latency. This means that they do not know when to execute a turn. When the two losses are combined, it is nearly impossible to fly the drone if the loss happens at the same time. This means the pilot has no idea where the drone is and cannot make a turn. In Tables 4 and 5, the loss in the case of Table 4 is higher, and this means that flying with video loss is more complicated. In the combined case (Table 6), the data show that the collision count does become greater after both losses are combined.

Moving forward, these simulated results show how critical communication loss is for a drone in a situation with many obstacles. The drone's path deviates more and more, and the collision probability increases. Similarly, since the controller of this simulator is just a simulated version of the same flight controller drones used in the industry, there is potential room for emulating the exact drone desired. This opens the door to even more realistic tests. The AirSim environment can also be modified to replicate the exact physical characteristics of a desired drone so the drone would react the same way as a desired drone, both in its controller and physical properties. This can also be seen by the results of Tables 1 and 2. As the number of walls increases, the packet loss increases, and the ability to fly the drone decreases (as seen between the good and poor results).

## 5. Conclusions

The simulations described in this paper were designed to help us understand drone behavior without potentially damaging flight testing. The main takeaway from these

tests is how drones flown by pilots are progressively more impacted by higher rates of communication loss with longer bursts of errors or packet loss. Our simulations showed that video loss affects piloting more than controller loss.

If the unit is moving straight, packet loss will not be critical, as when the unit needs to be precisely maneuvered, because units tend to act on the last received packet. Single packet loss more typical of an AWGN (Additive White Gaussian Noise) channel generally can be tolerated by the sUAS/Pilot system until the loss rate approaches 15% or more. Since the channel is not AWGN, it is characterized by a burst of errors due to multipath-induced fading. In a spatially constrained burst error environment, if the unit is close to a wall or needs to make frequent maneuvering, the likely outcome of a 0.5 to 1 s outage will be a collision with the wall. With enclosed propellers, the unit will likely survive; without the propellor guards, the unit will be severely damaged in a collision. "Sluggish controllability" often reported by pilots flying during the flight-testing phase was likely caused by a burst of errors on the control channel.

Having an early warning of loss of signal is critical in NLOS indoor/subterranean environments. The modulation and coding rapidly degraded over a few meters, and unless the operator could determine when to turn back, all the sUASs tested landed automatically and had to be manually recovered. In this case, the unit would be lost if operating in a hostile environment. In the ideal sense, the unit would be able to autonomously return to the starting point in a GPS-denied environment.

Since communication loss is common in underground environments, the simplest protection for the drone to continue operations would be having propeller guards. It will be very interesting if the exact drones are simulated with their flight controller in future tests. This does require a lot of proprietary information on the drones, which is not realistic for the Army sUAS being tested. But these experiments could be used by commercial drone makers to test their communication systems under various cases to determine how efficient their system is circumventing any interference. This could be a rigorous testbed for such future tests.

The final conclusion we drew from this effort is that the packet loss seen in flight testing can be modeled in the simulation. The increase in packet loss can be replicated by reducing the frequency and regularity of packets arriving. This can aid in the development of drones before building the prototype and reduce the cost of testing drones by letting them be tested in simulation and not damaged in real life.

**Author Contributions:** Conceptualization, E.M., J.W. and A.N.; methodology, E.M. and J.W.; software, E.M.; validation, E.M. and J.W.; formal analysis, E.M. and J.W.; investigation, E.M. and J.W.; resources, E.M. and A.N.; data curation, E.M.; writing—original draft preparation, E.M.; writing—review and editing, J.W.; visualization, E.M.; supervision, J.W.; project administration, A.N.; funding acquisition, A.N. All authors have read and agreed to the published version of the manuscript.

**Funding:** This project is sponsored by the Department of the Army, U.S. Army Combat Capabilities Development Command Soldier Center (W911QY-18-2-0006). Approved for public release: PAO# PR2023_98073.

**Data Availability Statement:** All the necessary data is contained within the article.

**Conflicts of Interest:** The authors declare no conflict of interest.

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
