# Peer review of "Simulation of the Effect of Correlated Packet Loss for sUAS Platforms Operating in Non-Line-of-Sight Indoor Environments"

_drones, doi:10.3390/drones7070485_

Round 1
Reviewer 1 Report
This paper examines how correlated packet loss (burst errors) occurring on both the control and camera communication links effects the ability of pilots to fly and navigate small sUAS platforms in constrained non-Line of Sight (NLOS) environments from a communication link point of view. This paper uses a UAV simulator, a test bench AirSim, to better understand the effects of correlated packet loss on flyability without damaging multiples UAS units by flight testing. However, there are still some suggestions and questions about this paper.
1. The innovation points of the paper are not very clear. Please provide a detailed list of the innovative work in Introduction.
2. The formatting of Table 4 appears to be disordered, and it is also split across multiple pages. I kindly request the authors to reorganize and adjust the layout of the table to ensure proper presentation and readability. It is important to maintain a clear and concise format for tables to enhance the overall quality of the manuscript.
3. In the section of Methodology for evaluating effect of packet loss on sUAS piloting, the authors gives four tables and a reference followed the four tables. Does these four tables summarized by the authours themselves or directly refered to Ref. [31]? Besides, the authors didn’t explain the table 3. Please give a detailed explanation of table 3.
4. The writing should be further improved and polished in this entire article, and there are some typos and grammar errors should be corrected.
5. The authors have designed two different test environments and used the AirSim platform to investigate the impact of correlated packet loss on the communication link of the sUAS platform. However, the main innovative aspects of their work are not clearly articulated. I kindly request the authors to provide a more detailed and explicit explanation of their primary contributions and innovations in this paper. This will help to enhance the clarity and significance of their research findings.
6. What research has been done on the impact of the drones or UAVs? Less has been written about the related work, and the authors' research lacks adequate literature review. Some refs could be useful, e.g., UAV-supported Intelligent Truth Discovery to Achieve Low-Cost Communications in Mobile Crowd Sensing. STMTO: A Smart and Trust Multi-UAV Task Offloading System. Towards minimum code dissemination delay through UAV joint vehicles for smart city
7. In the section of Methodology for evaluating effect of packet loss on sUAS piloting, the authors said “AirSim code was modified, and new routines developed to simulate the effects of correlated packet loss.”. Have the authors modified the AirSim code? if yes, please give the core code about this part.
Moderate editing of English language required
Author Response
Hello, thank you for the comments and the opportunity to make edits to this paper and resubmit. In regards to the comments made:
1. The goals and innovations has been re-explained in the introduction, methodology, results, and conclusion. The goal of the work is focused on developing simulated drones and make sure that their communication loss is properly modelled in simulation. This allows for drones to not be tested only in person where very expensive drones can be damaged.
2. Table 4 has been reformatted. But it is a large table and can be in its own page. All the parts of the table have been resized to reduce whitespace.
3. Table 1-4 have been condensed into 2. These tables are our previous work.
4.Typos and grammar reworked.
5. Goals and innovation has been re-explained (as in point 1).
6.We are confused how the papers listed might be helpful. But two more papers of background research were added in relationship to packet loss in drones
7.Airsim code modifications re-explained. UDP protocol is now explained. The communication layer over the Airsim code using ROS. This is done using ROS publishers and subscribers.
I hope this resolves any concerns you might have with this paper. Please let us know if anything else can be clarified or changed.
Reviewer 2 Report
Drones-2492864 Simulation of the Effect of Correlated Packet Loss for sUAS Platforms Operating in Non-Line-of-Sight Indoor
The paper examines how correlated packet loss occurring on both the control and camera communication links affects the ability of pilots to operate sUAS in a constrained non-Line of Sight (NLOS) environment. The presented idea is innovative and provides a practical and cost-effective solution to investigate this issue without actually damaging sUAS.
1. Line 83. Abbreviation needs an explanation when it first occurs, such as SNR.
2. Line 88 – 104. The font style needs formatting.
3. Tables 1 – 4 need to be simplified. Since you did not investigate the correlations between the quality of video latency and control communication versus the environment settings (distance/number of walls & floors), there is no need to present the full descriptions.
4. Figures 5 and 6, are there any observed correlations between the collision locations and the packet loss? This needs more investigation.
5. Collision count is used as the evaluation metric for the effect of correlated packet loss for sUAS operations in a Non-Line-of-Sight environment. Are there any other evaluation metrics? This needs more investigation.
6. Table 8 presents combined controller and video loss with different packet losses but needs more details. When I compare this to Tables 6 and 7, it seems that video loss is the predominant one. What are the percentages of controller and video loss, respectively, in Table 8? Are there any observed correlations regarding the rate (e.g., 10% of controller loss + 90 %of video loss versus 50% of controller loss + 50% of video loss versus 90% of controller loss + 10% of video loss) and the collision count?
7. Line 338 – 342. This paragraph is the template content and needs to be deleted.
Author Response
Hello, Thank you for the comments and the opportunity to make edits to this paper and resubmit. In regards to the comments made:
1. The SNR abbreviation has been explained.
2. The styling has been reformatted across the paper for consistency per the formatting for the Journal.
3.Tables one and two have been removed and merged effectively into tables 3 and 4 for simplicity. Also those tables have been reformatted to fit each in a single page.
4. The collision of the packet loss versus location relationship has been explained. It is primarily found near the corners because that is where the change in direction and the new command is needed. So a packet loss has most effect there if the new packet for the turn is lost.
5. The reasoning for the metrics has been re-explained. Collision is the only real metric because path deviation is not very useful when a person is flying. Even with no loss they cannot recreate the same path consistently. Effectively speaking when a drone moves in a straight line packet loss is not very dangerous. It is only important when packet loss occurs when a change in direction happens.
6. The fact that the main source of packet loss collision is the video feed loss. The proportions cannot be determined due to the fact this metric is not feasible. There is no way to test the exact percentage.
7. Deleted
I hope this resolves any concerns you might have with this paper. Please let us know if anything else can be clarified or changed.
Round 2
Reviewer 1 Report
The authors' efforts for improving the paper are sincerely appreciated. The paper is improved after the revisions. I have no additional comments.
The authors' efforts for improving the paper are sincerely appreciated. The paper is improved after the revisions. I have no additional comments.
Reviewer 2 Report
The authors have addressed my comments. No further revision is needed.